ⓐ | **Open Peer Review** | Clinical Microbiology | Observation

# Hundred-fold increase in SARS-CoV-2 spike antibody levels over three years in a hospital clinical laboratory

Patrizio Caturegli,[1] Oliver Laeyendecker,[2] Aaron A. R. Tobian,[1] David J. Sullivan[3]

**ABSTRACT** Natural infection with the SARS-CoV-2 virus and vaccination increase the serum spike antibody levels. These levels, which differ among the various commercial assays, are used by the FDA to qualify individuals as potential COVID-19 convalescent plasma (CCP) donors. Over a 3-year period (April 2020–February 2023), we performed a retrospective analysis of spike IgG antibodies measured by a tertiary hospital clinical immunology laboratory using the Euroimmun SARS-CoV-2 IgG ELISA. The 3-year interval was classified into five periods based on the SARS-CoV-2 strain variant epidemiology. A total of 15,820 sera, derived from 11,022 individuals (6,362 females and 4,660 males) ranging from severe immunocompromised state to routine health visits, were tested. Spike IgG levels rose significantly over time, from a median ELISA 0.13 Arbitrary Units (AU) in the first period to 48.7 in the last period ($P < 0.0001$). The 80th percentile of the spike IgG distribution was 0.55, 8.1, 9.6, 64.9, and 151 AU in the five periods. Using 3.5 AU, the FDA threshold for qualifying CCP donors with this Euroimmun assay, the percentage of subjects eligible for CCP donation would have been 11%, 44%, 61%, 81%, and 91% in the five time periods. Overall, spike antibody levels have risen more than 100-fold during the pandemic, while SARS-CoV-2 variants have become resistant to monoclonal antibodies. Since CCP containing high titers of spike antibodies is known to be most effective against variants, restricting CCP donors to those with antibody values in the upper two deciles may allow greater therapeutic transfusion protection.

**IMPORTANCE** Despite the evolution of SARS-CoV-2 variants of concern and ongoing transmission, COVID-19 hospitalization and mortality rates continue to decline. Both the percent seropositive and antibody levels have risen over the past 3 years. Here, we observe more than 90% seropositivity as well as more than a hundred-fold increase in spike IgG levels in a tertiary hospital clinical immunology laboratory setting. Antibody effector functions (such as neutralization, opsonization, and complement activation) and cell-mediated immunity all contribute to protection from COVID-19 progression to hospitalization, and all correlate to the total SARS-CoV-2 antibody levels. We recommend therapeutic COVID-19 convalescent plasma be restricted to the top 20% of potential donors to maintain activity against ongoing SARS-CoV-2 variant evolution.

**KEYWORDS** COVID-19, SARS-CoV-2, serologic population kinetics, plasma donor transfusions

Considering that populations have undergone multiple vaccinations and reinfections with evolving SARS-CoV-2 variants, we examined the validity of using 3.5 arbitrary units (AU) (in the original Euroimmun spike IgG antibody assay) as the threshold for selecting COVID-19 convalescent plasma (CCP) donors. We analyzed sera from patients seen at a tertiary hospital between 10 April 2020 and 28 February 2023 who had a spike antibody test ordered by their clinical provider. This 3-year time interval was classified into five periods to reflect the SARS-CoV-2 strain epidemiology: up to January 2021 for

Address correspondence to David J. Sullivan, dsulliv7@jhmi.edu.

The authors declare no conflict of interest.

See the funding table on p. 5.

the original strain; January to June 2021 for the alpha variants with partial vaccinations; July to November 2021 for the delta variants; December 2021 to June 2022 for the omicron BA.1 and 2; and July 2022 to February 2023 for the omicron BA.4/5 with BQ.1 and XBB.

A total of 11,022 subjects, 6,362 females and 4,660 males, from hospitalized immunocompromised to vaccinated immunocompetent outpatient individuals, contributed 15,820 serum samples measured for spike IgG antibodies. The age was similar in the two genders across the five time periods (Table S1), with an overall mean [standard deviation (SD)] of 50 (20) years. Females had overall significantly higher antibody levels than males (Table S1), in keeping with the notion that B cell number and serum antibody levels are greater in adult women (1). SARS-CoV-2 antibodies increased significantly over time, from a median 0.13 in the pre-vaccination period to 1.85, 5.25, 11.7, and 48.8 AU in the following periods (Fig. 1). Adjusting for gender, each period had significantly higher spike IgG levels ($P < 0.0001$) than the preceding period (Table S2). Most sera (2,763 of 3,109, 89%) tested negative during the first period, while most tested positive (2,306 of 2,422, 95%) in the last one. For context in Maryland, the cumulative cases, vaccination percentages, seropositive rates from CDC hosted SeroNet and CDC national blood donor database are tabulated along the seroprevalence in the time periods of this study (Table S3). The 80th percentile of the spike IgG distribution in the five time periods was 0.55, 8.1, 9.6, 64.9, and 151 AU (150 AU ~ 2,100 RU/mL ~ 6,720 BAU/mL), indicating a greater than 250-fold increase when comparing last to first periods. The increasing trend in spike antibody levels was confirmed when longitudinal data analysis was performed in the subset of subjects (2,571 of 11,022, 23%) who were tested longitudinally two or more times (Fig. 2A). Using the FDA CCP eligibility threshold of 3.5 AU would have classified eligible CCP donors, 346 of the 3,109 sera (11%) in period 1; 1,324 of 3,039 (44%) in period 2; 2,263 of 3,724 (61%) in period 3; 2,849 of 3,526 (81%) in period 4; and 2,213 of 2,422 (91%) in period 5. Thus, the vast majority of individuals currently qualify as potential CCP donors since they have spike antibody levels exceeding the FDA criteria for "high titer" CCP.

The newer Euroimmun spike IgG assay features a standard curve composed of six calibrators, rather than the single calibrator found in the original assay. Although the newer assay was not used for routine patient testing, we compared the two assays in a subset of 552 patient sera. The assays showed a highly significant linear correlation (adjusted $r$-squared of 0.861, $P < 0.0001$, Fig. 2B): for every unit increase in the original assay optical density ratio, the newer assay increased 14 RU/mL (95% CI = 13.1–15.7) units. Of the 417 individuals compared in June 2021, 243 (58%) would have qualified as CCP donors having spike antibody levels >3.5 AU in the original assay and >55 RU/mL in the newer assay. On the contrary, all 135 patients tested in February 2023 would meet the qualification criteria.

This study demonstrated that most individuals now qualify as a CCP donor. During the initial COVID-19 pandemic, therapeutic CCP was selected based on the donor's symptomatology and any positive SARS-CoV-2 laboratory test (2). By February 2021, the FDA established therapeutic threshold values for the most commonly used serum spike antibody assays (3), such as the 3.5 AU for the original Euroimmun assay (approved for the US market on 4 May, 2020), or the nearly equivalent (since 3.5 times 14 equals 49 RU/mL) 55 RU/mL units for the newer Euroimmun assay ("Anti-SARS-CoV-2 Curve ELISA," approved on 5 October, 2021) (4). In the first period, only 11% of subjects tested in the clinical lab had a value greater than 3.5 AU. Since ideal donors for therapeutic CCP are those with the highest spike antibody levels, we suggest increasing the threshold as to include only those who have antibodies in the upper two deciles of the spike antibody distribution. Vaccine efficacy metrics often uses the reference of "COVID-19 convalescent plasma" levels (5, 6), which is a broad range as seen here. We advocate that CCP units used for therapy comprise the upper deciles as the goal at present and in the future for therapeutic CCP. The volume of distribution for CCP approaches 3–5 L with 15- to 20-fold dilution from 250 mL of plasma (7). An outpatient treatment CCP study qualified the top

| Fold increase over baseline: | | 15 | 17 | 118 | 275 |
|---|---|---|---|---|---|
| 80th percentile: | 0.55 | 8.1 | 9.6 | 64.9 | 151 |
| Percent > 3.5 | 11% | 44% | 61% | 81% | 91% |
| No. of sera: | 3,109 | 3,039 | 3,724 | 3,526 | 2,422 |

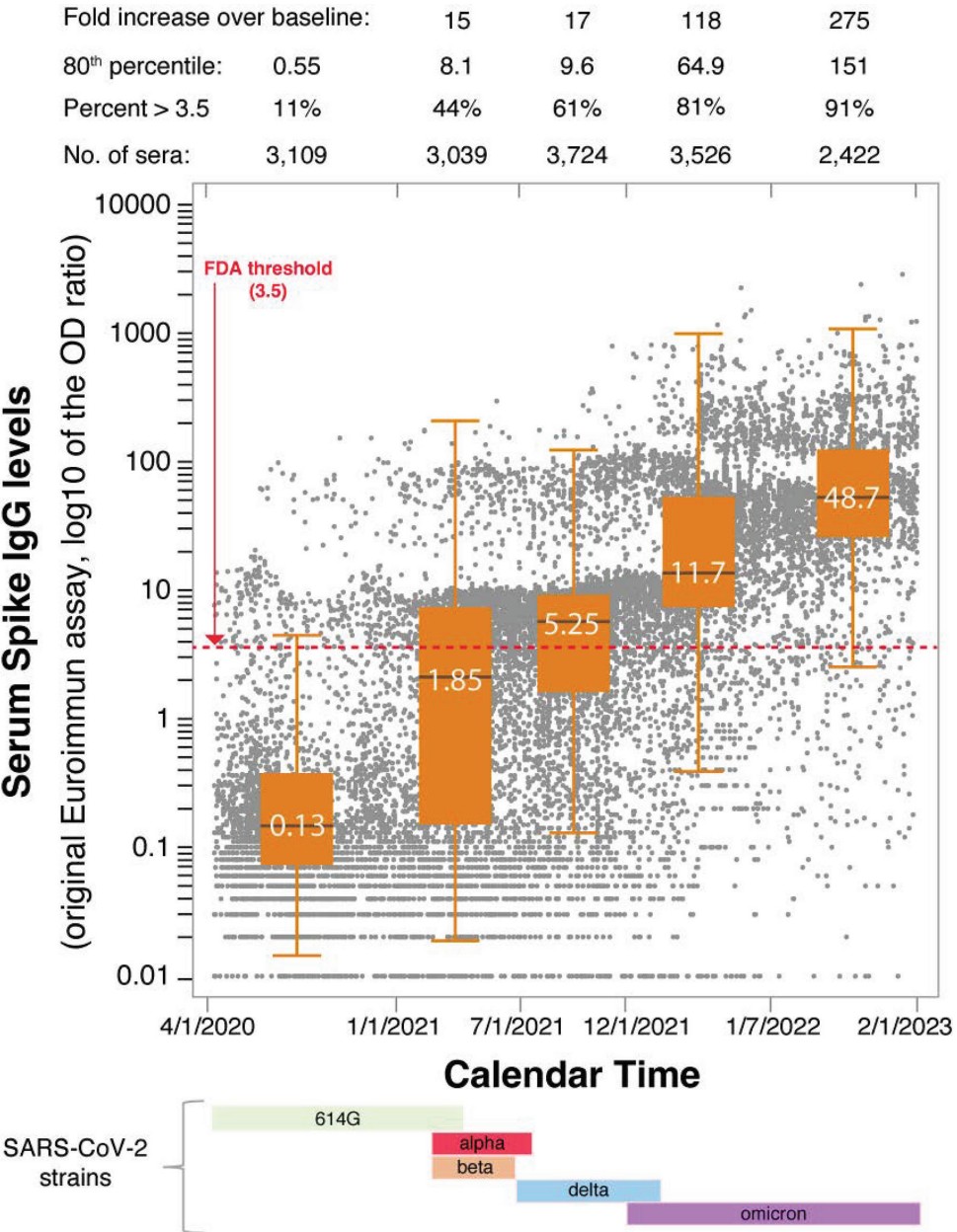

**FIG 1** Increasing trend of spike IgG antibodies in a tertiary hospital patient population. The 3-year period between April 2020 and February 2023 was divided into five time periods to represent the alpha variant (up to January 2021), the alpha variants with partial vaccinations (February to June 2021), the delta variants (July to December 2021), the omicron BA.1 and 2 (January to July 2022), and the omicron BA.4/5 with BQ.1 and XBB variants (August to February 2023). A total of 15,820 sera are shown by the individual points, contributed by 11,022 patients. Most patients (8,451, 77%) were tested only once during the 3 years, while the remaining 2,571 underwent sequential measurements. The box plot in each of the five time periods represents the middle 50% of the observations, bordered at the 25th and 75th percentiles, and contain a line indicating the median antibody value. The 3.5 ELISA optical density value suggested by the FDA as cutoff for COVID-19 convalescent plasma donations is shown by the red dashed line. Presently, high antibody level patient samples require a 1–5,000 serum dilution in contrast to 1:101 suggested in the original Euroimmun. The conversion of Euroimmun AU 1 = 14 RU/mL = 45 BAU/mL such that on the low range 3.5 AU ~ 49 RU/mL ~ 157 BAU/mL and on the high range 150 AU ~ 2,100 RU/mL ~ 6,720 BAU/mL.

60% of donors and demonstrated in the top 30%, early plasma administration reduced hospital risk 92% (95% CI 41%–99%, $P$ = 0.014) (8). The high titer plasma deciles retain potent virus neutralization against current and future variants for months (9–11). There

**A**

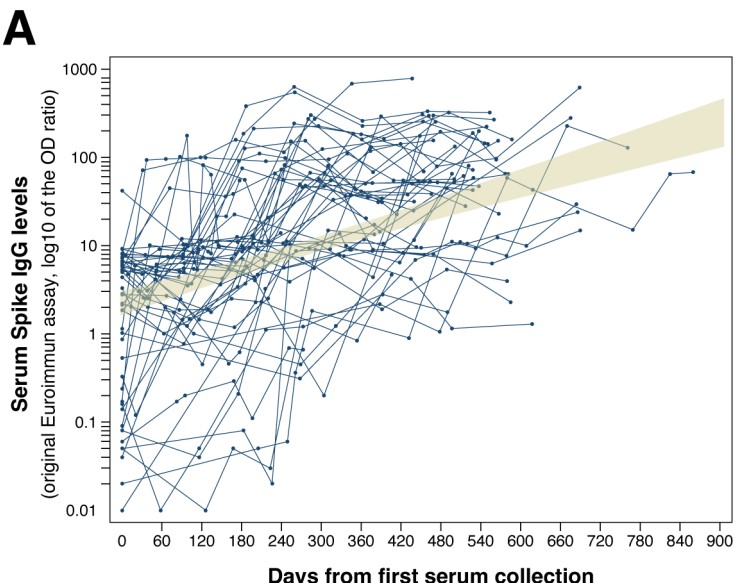

**B**

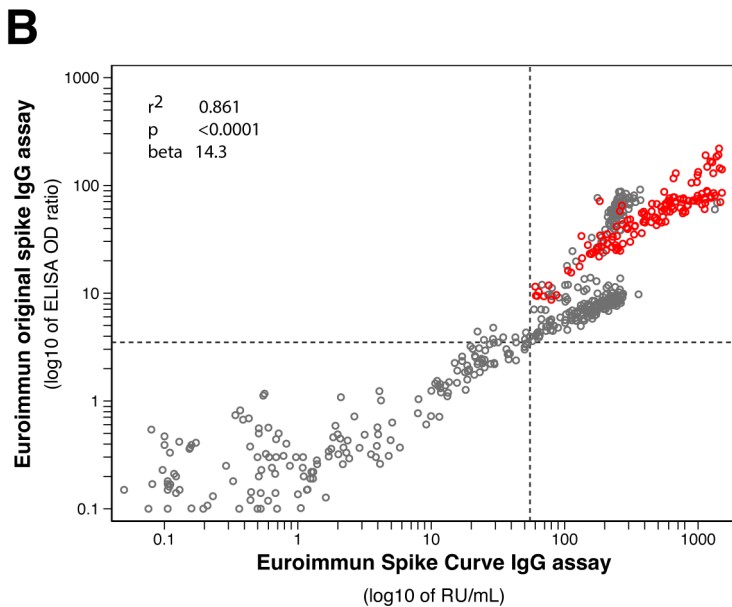

**FIG 2** (A) Sequential spike IgG antibody levels in 49 of the total 2,571 patients with sequential measurements. The 49 patients were tested between 7 and 13 times, contributed 381 sera, and had a mean follow-up time of 381 days. The shaded area represents the 95% confidence interval around the linear fit (which is not shown). (B) Comparison of the two Euroimmun spike IgG antibody ELISA assays, the original version released in May 2020 and the newer version with a standard curve released in October 2021. The dotted lines indicated the FDA recommended threshold for COVID-19 convalescent plasma donation: 3.5 AU for the original assay on y-axis and 55 RU/mL for the newer assay on x-axis. The comparison was made in June 2021 for the gray and February 2023 for the red symbols. The conversion of Euroimmun AU 1 = 14 RU/mL = 45 BAU/mL such that on the low range 3.5 AU ~ 49 RU/mL ~ 157 BAU/mL and on the high range 150 AU ~ 2,100 RU/mL ~ 6,720 BAU/mL.

are early data from vaccination cohorts, that despite the fold drop in virus neutralizations by XBB.* and BQ.1, hospital rates are still low (CDC Tracker 12 February 2023 at 1.1

hospitalizations per 100,000 falling from 2/100,000 on 5 January 2023) with current vaccinations.

As monoclonal antibody therapy has become ineffective, there is increased interest in polyclonal CCP to complement small-molecule antiviral drug therapy, especially for the immunocompromised patients and those at highest risk of hospitalization (12, 13). There have been offsetting trends. On one hand, with more vaccine boosters and cumulative COVID-19 incidence, the levels of neutralizing antibodies (not the total spike antibody measured here) are trending to 10 times the geometric means from the original CCP from unboosted WA-1 or pre-alpha COVID-19 (9, 10). On the other hand, new Omicron variants like XBB and BQ are more than 10 times resistant to virus neutralization compared to WA-1 with preBQ or preXBB vaccine AND recent Omicron plasma (11). The present blood donor qualification system in the past has tolerated the rough correlation of total spike, S-1, or RBD antibodies to virus neutralization (14, 15).

A strength of this analysis is the more than 15,000 samples over a 3-year period on a single Euroimmun serologic assay platform. Limitations include variable number of immunocompromised antibody deficient individuals and consolidation of data points near 10 and 100 AU from not fully diluting all the samples (up to 1:5,000 presently) both which may underestimate population increases in antibody levels.

A conservative projection is that the 10- to 100-fold increase in SARS-CoV-2 antibody levels seen over the past 3 years will begin to plateau in the next years which predict less change of antibody levels (Fig. S1). For ongoing use in the immunocompromised patients lacking sufficient SARS-CoV-2 antibodies (13), the upper deciles of available CCP donor units will provide the highest effective viral specific antibody dose for the longest duration against both matched and mismatched SARS-CoV-2 variants.

## ACKNOWLEDGMENTS

Support was provided, in part, by the Division of Intramural Research, NIAID, NIH, and Bloomberg Family Foundation.

D.J.S. and P.C. designed data analysis from existing hospital data. P.C. analyzed data, and D.J.S. drafted the manuscript with input from all authors, who then approved the final version of the manuscript.

All authors report no relevant disclosures.

## AUTHOR AFFILIATIONS

[1]Department of Pathology, Johns Hopkins University School of Medicine, Johns Hopkins University, Baltimore, Maryland, USA

[2]Division of Intramural Research, National Institute of Allergy and Infectious Diseases, NIH, Bethesda, Maryland, USA

[3]Department of Molecular Microbiology and Immunology, Johns Hopkins Bloomberg School of Public Health, Johns Hopkins University, Baltimore, Maryland, USA

## AUTHOR ORCIDs

Oliver Laeyendecker https://orcid.org/0000-0002-6429-4760
Aaron A. R. Tobian http://orcid.org/0000-0002-0517-3766
David J. Sullivan http://orcid.org/0000-0003-0319-0578

## FUNDING

| Funder | Grant(s) | Author(s) |
| --- | --- | --- |
| Bloomberg Family Foundation (The Bloomberg Family Foundation) | | David J. Sullivan |
| HHS | NIH | NIAID | Division of Intramural Research, National Institute of Allergy and Infectious Diseases (DIR, NIAID) | | Oliver Laeyendecker |

## AUTHOR CONTRIBUTIONS

Patrizio Caturegli, Conceptualization, Data curation, Formal analysis, Methodology, Project administration, Supervision, Validation, Visualization, Writing – review and editing | Oliver Laeyendecker, Data curation, Methodology, Visualization, Writing – review and editing | Aaron A. R. Tobian, Conceptualization, Formal analysis, Methodology, Visualization, Writing – review and editing | David J. Sullivan, Conceptualization, Data curation, Formal analysis, Funding acquisition, Methodology, Visualization, Writing – original draft, Writing – review and editing

## ADDITIONAL FILES

The following material is available online.

### Supplemental Material

**Supplemental material (Spectrum02183-23-s0001.docx).** Tables S1 to S3 and Fig. S1.

### Open Peer Review

**PEER REVIEW HISTORY (review-history.pdf).** An accounting of the reviewer comments and feedback.

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
