## [Reviewer comments · Microbiology Spectrum]

Microbiology Spectrum

Hundred-fold increase in SARS-CoV-2 spike antibody levels over three years in a hospital clinical laboratory.

Patrizio Caturegli, Oliver Laeyendecker, Aaron Tobian, and David Sullivan

Corresponding Author(s): David Sullivan, Johns Hopkins University Bloomberg School of Public Health

Review Timeline:

Submission Date:	May 24, 2023
Editorial Decision:	August 15, 2023
Revision Received:	August 22, 2023
Accepted:	August 29, 2023

Editor: Bar-On Yotam

Reviewer(s): The reviewers have opted to remain anonymous.

Transaction Report:

DOI: <https://doi.org/10.1128/spectrum.02183-23>

August 15, 2023

Dr. David J Sullivan
Johns Hopkins University Bloomberg School of Public Health
Molecular Microbiology & Immunology
615 N. Wolfe St., Rm. W 4606
Baltimore, MD 21205

Re: Spectrum02183-23 (Hundred-fold increase in SARS-CoV-2 spike antibody levels over three years in a hospital clinical laboratory.)

Dear Dr. David J Sullivan:

Minor modifications are required before publication.
Please correct the manuscript according to the suggestions of reviewers 1 and 2.

Link Not Available

Sincerely,

Bar-On Yotam

Journals Department
Reviewer comments:

Reviewer #1 (Comments for the Author):

This is an interesting and significant descriptive study of the changes in the level of antibodies against spikes throughout the evolution of the COVID-19 pandemic. The authors, through a simple analytical approach, attempt to support the use of CCP donors as an anti-SARS-CoV-2 treatment, in particular for immunocompromised patients.

Reviewer #2 (Comments for the Author):

I appreciate the opportunity to review this interesting manuscript by Caturegli et al.

The authors analyzed spike antibodies measured by a tertiary hospital clinical immunology laboratory over a 3-year period, (April 2020 - February 2023), using the dilutional "Anti-SARS-CoV-2 ELISA" assay (Euroimmun US, Mountain Lakes, NJ). The study showed that spike IgG levels rose markedly over time, from a median of 0.13 ELISA arbitrary units (AU) in period 1 to 48.7 in period 5 ($p < 0.0001$). The spike IgG 80th percentile distribution threshold was 0.55, 8.1, 9.6, 64.9, and 151 AU in each of these periods.

This manuscript is an interesting piece of work, but it is not in a final state to be published yet. Therefore, the manuscript requires corrections before considering its acceptance.

Comments:

The abstract is well-written and informative. It clearly states the purpose of the study, the methods used, the main findings, and the implications of the findings.

Methods: the study is well-designed. The authors used a large sample size and a validated assay.

The findings of the study are important and have implications for the use of COVID-19 convalescent plasma therapy.

The authors mention that the FDA uses a threshold of 3.5 AU to qualify CCP donors. However, they do not discuss why this threshold was chosen and why the 55 AU was also chosen in the newer assay.

The authors suggested that restricting CCP donors to those with high titre spike antibodies may be more effective in protecting immunocompromised patients from variants. However, they do not provide any data to support this suggestion. It would be helpful to see data on the clinical effectiveness of CCP therapy in patients with different levels of spike antibodies.

Other comments are highlighted in the attached Pdf.

Staff Comments:

Preparing Revision Guidelines

Please return the manuscript within 60 days; if you cannot complete the modification within this time period, please contact me. If you do not wish to modify the manuscript and prefer to submit it to another journal, please notify me of your decision immediately so that the manuscript may be formally withdrawn from consideration by Microbiology Spectrum.

Hundred-fold increase in SARS-CoV-2 spike antibody levels over three years in a hospital clinical laboratory.

In this study, the authors evaluated the increase in neutralizing antibodies against SARS-CoV-2 over a three-year period in the setting of a clinical study. This time period was divided into five intervals defined by the U.S. population prevalence of major SARS-CoV-2 mutations. The authors found an increase in antibody levels against the RBD region of the spike. The increase was even more evident when the 80th percentile of the five intervals were compared. Based on these results, the authors suggest redefining threshold levels of antibodies that could be used to identify COVID-19 convalescent plasma (CCP) donors.

This is an interesting and significant descriptive study of the changes in the level of antibodies against spikes throughout the evolution of the COVID-19 pandemic. The authors, through a simple analytical approach, attempt to support the use of CCP donors as an anti-SARS-CoV-2 treatment, in particular for immunocompromised patients. A major limitation of the study is the lack of a neutralization test. These tests could have been performed on a subset of data from each of the time intervals. A correlation between antibody levels and neutralization was previously demonstrated [1]. The present study could have determined the correlation between the neutralization tests and the categorization that is proposed based on the level of antibodies. Or the authors could have used the results of the two different tests to infer the CCP donor selection thresholds. Although the authors' proposal to use the upper 20% of the distribution of antibody levels is consistent with what was previously suggested by other authors [2], this categorization is not supported from a quantitative point of view. A second point that could have enriched this work would have been the identification of variables associated with the number of antibodies. Indeed, given the large volume of data, it would have been interesting to identify clinical, epidemiological, or biological variables of the virus that are associated with the number of antibodies. This exploratory analysis could identify criteria to take into account in the definition of thresholds for the selection of CCP donors.

Suggestions

- The authors could add a table (plot) describing the population analyzed. This could be organized by time intervals; in each of them, it could be indicated, for example, the number of women and men, the number of people vaccinated, the number of people with previous SARS-CoV-2 infections, and the number of seropositive patients. This table could be cited in the text.
- In addition to the median, it can be informative to add to the text the minimum and maximum values of the number of antibodies.
- The authors could evaluate if there are significant statistical differences between the median number of antibodies of the different time intervals.
- The authors could evaluate if there are significant statistical differences between the median of seropositives of the different time intervals.
- The authors could evaluate if there are significant statistical differences in the number of antibodies between sexes for the same time interval or between different intervals for the same sex.

The results of these tests could support some of the claims made in the text.

References

1. <https://doi.org/10.1128/mbio.03523-22>
2. <https://doi.org/10.1371/journal.pone.0273223>

**Observation**

Hundred-fold increase in SARS-CoV-2 spike antibody levels over three years in a hospital clinical
laboratory.

Running title- Hundred-fold increase in spike antibody levels

¹Patrizio Caturegli, ²Oliver Laeyendecker, ¹Aaron A.R. Tobian, ⁴David J. Sullivan

¹Department of Pathology, Johns Hopkins University School of Medicine, Johns Hopkins University,
Baltimore, MD

²Division of Intramural Research, National Institute of Allergy and Infectious Diseases, NIH

³Department of Molecular Microbiology and Immunology, Johns Hopkins Bloomberg School of
Public Health, Johns Hopkins University, Baltimore, MD

Correspondence: David Sullivan, Rm W4606, 615 N. Wolfe St, Baltimore, MD 21205,

dsulliv7@jhmi.edu, telephone-410 502 2522 fax-410 955 0105

**Funding-** none

Abstract 249 (limit 250 words)

Word count is 1139

Figures 2 Tables 0

References 17

Conflict of interest- Authors declare no competing financial interests

**Abstract**

Natural infection with the SARS-CoV-2 virus and spike protein immunization increase the serum spike
antibody levels. These levels, which differ among the various commercial assays, are used by the FDA
to qualify individuals as potential COVID-19 convalescent plasma (CCP) donors. Over a 3-year
period, (April 2020 – February 2023), we analyzed spike antibodies measured by a tertiary hospital
clinical immunology laboratory using the dilutional “Anti-SARS-CoV-2 ELISA” assay (Euroimmun
US, Mountain Lakes, NJ). The three year interval was arbitrarily classified into five periods based on
the SARS-CoV-2 strain variant epidemiology. A total of 15,820 sera, derived from 11,022 individuals
(6,362 females, mean age 50±21 years), ranging from severe immunocompromised state to routine
health visits, were tested for spike IgG antibodies. Spike IgG levels rose markedly over time, from a
median of 0.13 ELISA arbitrary units (AU) in period 1 to 48.7 in period 5 (p<0.0001). The spike IgG
80th percentile distribution threshold was 0.55, 8.1, 9.6, 64.9, and 151 AU in each of these periods.
Using the 3.5 AU threshold the FDA uses to qualify CCP donors with this assay, the percentage of
subjects eligible for CCP donation would have been 11%, 44%, 61%, 81%, and 91% in the five time
periods. Antibody levels have risen more than hundred-fold while variants have become resistant to
clinically available monoclonal antibodies. As high-titer CCP is most effective against variants,
restricting CCP donors to those with spike antibody levels in the upper two deciles may allow
protection against variants when transfused to the immunocompromised.

Importance-112 words (150 limit)
Despite the evolution of SARS-CoV-2 variants of concern and ongoing transmission, COVID-19
hospitalization and mortality rates continue to decline. Both the percent seropositive and antibody
levels have risen over the past three years. Here we observe more than 90% seropositivity as well as
more than a hundred-fold increase in spike IgG levels in a tertiary hospital clinical immunology
laboratory setting. Antibody effector functions (such as neutralization, opsonization, and complement
activation) and cell mediated immunity all contribute to protection from COVID-19 progression to
hospitalization, and all correlate to the total SARS-CoV-2 antibody levels. We recommend therapeutic
COVID-19 convalescent plasma be restricted to the top 20% of potential donors to maintain activity
against ongoing SARS-CoV-2 variant evolution.

Keywords

COVID-19, SARS-CoV-2, serologic population kinetics, plasma donor transfusions

Considering that populations have undergone multiple vaccinations and reinfections with
evolving SARS-CoV-2 variants, we analyzed the validity of the Euroimmun 3.5 arbitrary unit (AU)
threshold using a patient cohort of clinical provider ordered serology, seen at a tertiary hospital
between April 10, 2020 and Feb 28, 2023. This 3-year time interval was classified into five periods to
reflect the epidemiology of the SARS-CoV-2 strains: up to January 2021 for the original strain;
January to June 2021 for the alpha variants with partial vaccinations; July to November 2021 for the
delta variants; December 2021 to June 2022 for the omicron BA.1 and 2; and July 2022 to January
2023 for the omicron BA.4/5 with BQ.1 and XBB.

A total of 11,022 subjects (6,362 females and 4,660 males, 50±21 years of age) contributed
15,820 samples measured for spike IgG antibodies. The study population ranged from
immunocompromised to vaccinated immunocompetent individuals. SARS-CoV-2 antibody levels
increased significantly over time, from a median AU of 0.13 in the pre-vaccination period to 1.8, 5.3,
11.7, and 48.8 AU in the following periods ($p < 0.0001$, Figure 1). During the first period, most sera
(2,763 of 3,109, 89%) tested negative (i.e., value < 1.23 AU), while during the last period most tested
positive (2,306 of 2,422, 95%). The 80th percentile of the spike IgG distribution was 0.55, 8.1, 9.6,
64.9, and 151 AU in the five time periods, indicating a greater than 250-fold increase when comparing
last to first periods. The increasing trend in spike antibody levels was confirmed when longitudinal
data analysis was performed in the subset of subjects (2,571 of 11,022, 23%) who were tested two or
more times (Figure 2A). Using the FDA COVID-19 convalescent plasma (CCP) eligibility threshold of
3.5 AU would have classified eligible CCP donors, 346 of the 3109 sera (11%) in period 1, 1324 of
3039 (44%) in period 2, 2263 of 3724 (61%) in period 3, 2849 of 3526 (81%) in period 4, and 2213 of
2422 (91%) in period 5. Thus, the vast majority of individuals currently qualify as potential CCP
donors since they have spike antibody levels exceeding the FDA criteria for “high titer” CCP.

The newer Euroimmun spike IgG assay features a standard curve composed of 6 calibrators
(ranging in concentration from 1 to 120 RU/mL), rather than the single calibrator found in the original

assay. Although the newer assay was not used in this study, we compared the two assays in a subset of
subjects (552 of 11,022, 5%). Upon transforming the raw antibody results to the log₁₀ scale, the two
assays showed a highly significant linear correlation (adjusted r-squared of 0.861, p<0.0001, Figure
2B): for every unit increase in the value of the original assay, the value of the newer assay increased 14
RU/mL (95% CI from 13.1 to 15.7). A total of 58% of the subjects (243 of 417) evaluated for the assay
comparison in June 2021 (gray symbols) would qualify as CCP donors because having spike antibody
levels >3.5 AU in the original assay and >55 RU/mL in the newer assay. On the contrary, all 135
random patient subjects tested in February 2023 (red symbols) would meet the qualification criteria.

This study demonstrated that most individuals now qualify as a potential CCP donor. During
the initial COVID-19 pandemic months, therapeutic CCP was selected based on the donor's
symptomatology and any positive SARS-CoV-2 laboratory test(1). By February 2021, the FDA had
established therapeutic threshold values for the most commonly used serum spike antibody assays(2),
such as the 3.5 AU for the original Euroimmun "Anti-SARS-CoV-2 ELISA" assay (approved for the
US market on May 4, 2020), or the near equivalent threshold 55 RU/mL for the later version ("Anti-
SARS-CoV-2 Curve ELISA", approved on October 5, 2021)(3). Since ideal donors for therapeutic
CCP are those with the highest spike antibody levels, we suggest increasing the threshold as to include
only those who have antibodies in the upper two deciles of the spike antibody distribution. Vaccine
efficacy metrics often uses the reference of "COVID-19 convalescent plasma" levels(4, 5), which is a
broad range as seen here. We advocate that CCP units used for therapy comprise the upper quintile as
the goal at present and in the future for therapeutic CCP. The volume of distribution for CCP
approaches 3-5 L with 15-to-20-fold dilution from 250 milliliters of plasma(6). An outpatient treatment
CCP study qualified the top 60% of donors and demonstrated in the top 30%, early plasma
administration reduced hospital risk 92%(95% CI 41%-99%, p=.014)(7). The high titer plasma quintile
retains potent virus neutralizations against current and future variants for months(8, 9).

As monoclonal antibody therapy has become ineffective, there is increased interest in
polyclonal CCP to complement small molecule antiviral drug therapy, especially for the
immunocompromised patients and those at highest risk of hospitalization(10, 11). There have been
offsetting trends. On one hand with more vaccine boosters and cumulative COVID-19 incidence, the
levels of neutralizing antibodies (not the total spike antibody measured here) are trending to ten times
the geometric means from the original CCP from unboosted WA-1 or pre-alpha COVID-19(9, 12). On
the other hand new Omicron variants like XBB and BQ are more than ten times resistant to virus
neutralization compared to WA-1 with preBQ or preXBB vaccine AND recent Omicron plasma(13).
The present blood donor qualification system in the past has tolerated the rough correlation of total
spike, S-1 or RBD antibodies to virus neutralization(14, 15).

True identical match to circulating variants like XBB with collection and rapid dispensing may
not be achievable. However, primary virus neutralization data and two systemic reviews show CCP
from clinical cohorts (not qualified donor units) collected after previous variants are still able to
neutralize both existing and future variants(12, 13). High titer viral mismatch neutralizes to the same
extent as medium titer match. There is early data from vaccination cohorts, that despite the fold drop in
virus neutralizations by XBB.* and BQ.1, hospital rates are still low (CDC Tracker Feb 12, 2023 at
1.1 hospitalizations per 100,000 falling from 2/100,000 on Jan 5, 2023) with current vaccinations.

A strength of this analysis is the more than 15 thousand samples over a three-year period on a
single Euroimmun serologic assay platform. Limitations include variable number of
immunocompromised antibody deficient individuals and consolidation of data points near 10 and 100
AU from not fully diluting all the samples both which may underestimate population increases in
antibody levels.

A conservative projection is that the 10 to 100 fold increase in SARS-CoV-2 antibody levels
seen over the past 3 years will begin to plateau in the next years which predicts less change of antibody
levels. For ongoing use in the immunocompromised patients lacking sufficient SARS-Cov-2

antibodies(16), the upper quintile of available CCP donor units will provide the highest effective viral
specific antibody dose for the longest duration against both matched and mismatched SARS-CoV-2
variants.

**References**

- 1. Tobian AAR, Cohn CS, Shaz BH. 2022. COVID-19 convalescent plasma. *Blood* 140:196-207.
- 2. Villa C. 2021. EUA 26382 COVID-19 Convalescent Plasma. OBRR, FDA
(<https://www.fda.gov/media/155159/download>).
- 3. Caturegli G, Materi J, Howard BM, Caturegli P. 2020. Clinical Validity of Serum Antibodies to
SARS-CoV-2 : A Case-Control Study. *Ann Intern Med* 173:614-622.
- 4. Goldblatt D, Alter G, Crotty S, Plotkin SA. 2022. Correlates of protection against SARS-CoV-
2 infection and COVID-19 disease. *Immunol Rev* 310:6-26.
- 5. Goldblatt D, Fiore-Gartland A, Johnson M, Hunt A, Bengt C, Zavadska D, Snipe HD, Brown
JS, Workman L, Zar HJ, Montefiori D, Shen X, Dull P, Plotkin S, Siber G, Ambrosino D.
2022. Towards a population-based threshold of protection for COVID-19 vaccines. *Vaccine*
40:306-315.
- 6. Shoham S, Bloch EM, Casadevall A, Hanley D, Lau B, Gebo K, Cachay E, Kassaye SG,
Paxton JH, Gerber J, Levine AC, Naeim A, Currier J, Patel B, Allen ES, Anjan S, Appel L,
Baksh S, Blair PW, Bowen A, Broderick P, Caputo CA, Cluzet V, Elena MC, Cruser D,
Ehrhardt S, Forthal D, Fukuta Y, Gawad AL, Gniadek T, Hammel J, Huaman MA, Jabs DA,
Jedlicka A, Karlen N, Klein S, Laeyendecker O, Karen L, McBee N, Meisenberg B, Merlo C,
Mosnaim G, Park HS, Pekosz A, Petrini J, Rausch W, Shade DM, Shapiro JR, Singleton RJ,
Sutcliffe C, et al. 2022. Transfusing convalescent plasma as post-exposure prophylaxis against
SARS-CoV-2 infection: a double-blinded, phase 2 randomized, controlled trial. *Clin Infect Dis*
doi:10.1093/cid/ciac372.
- 7. Levine AC, Fukuta Y, Huaman MA, Ou J, Meisenberg BR, Patel B, Paxton JH, Hanley DF,
Rijnders BJ, Gharbharan A, Rokx C, Zwaginga JJ, Alemany A, Mitjà O, Ouchi D, Millat-
Martinez P, Durkalski-Mauldin V, Korley FK, Dumont LJ, Callaway CW, Libster R, Marc GP,
Wappner D, Esteban I, Polack F, Sullivan DJ. 2023. COVID-19 Convalescent Plasma

- Outpatient Therapy to Prevent Outpatient Hospitalization: A Meta-analysis of Individual
Participant Data From Five Randomized Trials. *Clinical Infectious Diseases*
doi:10.1093/cid/ciad088.
- 8. Sullivan DJ, Franchini M, Joyner MJ, Casadevall A, Focosi D. 2022. Analysis of anti-SARS-
CoV-2 Omicron-neutralizing antibody titers in different vaccinated and unvaccinated
convalescent plasma sources. *Nature Communications* 13:6478.
- 9. Li M, Beck EJ, Laeyendecker O, Eby YJ, Tobian AA, Caturegli P, Wouters C, Chiklis G,
Block W, McKie R, Joyner M, Wiltshire TD, Dietz AB, Gniadek TJ, Shapiro A, Yarava A,
Lane K, Hanley DF, Bloch EMM, Shoham S, Cachay E, Meisenberg BR, Huaman M, Fukuta
Y, Patel B, Heath SL, Levine AC, Paxton JH, Shweta A, Gerber J, Gebo KA, Casadevall A,
Pekosz A, Sullivan DJ, Consortium C-SS. 2022. Convalescent plasma with a high level of
virus-specific antibody effectively neutralizes SARS-CoV-2 variants of concern. *Blood Adv*
doi:10.1182/bloodadvances.2022007410:2022.03.01.22271662.
- 10. Estcourt LJ, Cohn CS, Pagano MB, Iannizzi C, Kreuzberger N, Skoetz N, Allen ES, Bloch EM,
Beaudoin G, Casadevall A, Devine DV, Foroutan F, Gniadek TJ, Goel R, Gorlin J, Grossman
BJ, Joyner MJ, Metcalf RA, Raval JS, Rice TW, Shaz BH, Vassallo RR, Winters JL, Tobian
AAR. 2022. Clinical Practice Guidelines From the Association for the Advancement of Blood
and Biotherapies (AABB): COVID-19 Convalescent Plasma. *Ann Intern Med* 175:1310-1321.
- 11. Bloch EM, Focosi D, Shoham S, Senefeld J, Tobian AAR, Baden LR, Tiberghien P, Sullivan
D, Cohn C, Dioverti V, Henderson JP, So-Osman C, Juskewitch JE, Razonable RR, Franchini
192 M, Goel R, Grossman BJ, Casadevall A, Joyner MJ, Avery RK, Pirofski LA, Gebo K. 2023.
Guidance on the use of convalescent plasma to treat immunocompromised patients with
COVID-19. *Clin Infect Dis* doi:10.1093/cid/ciad066.

- 12. Sullivan DJ, Franchini M, Joyner MJ, Casadevall A, Focosi D. 2022. Analysis of anti-SARS-
CoV-2 Omicron-neutralizing antibody titers in different vaccinated and unvaccinated
convalescent plasma sources. *Nat Commun* 13:6478.
- 13. Sullivan DJ, Franchini M, Senefeld JW, Joyner MJ, Casadevall A, Focosi D. 2022. Plasma
after both SARS-CoV-2 boosted vaccination and COVID-19 potentially neutralizes BQ.1.1 and
XBB.1. *bioRxiv* doi:10.1101/2022.11.25.517977.
- 14. Patel EU, Bloch EM, Clarke W, Hsieh YH, Boon D, Eby Y, Fernandez RE, Baker OR, Keruly
202 M, Kirby CS, Klock E, Littlefield K, Miller J, Schmidt HA, Sullivan P, Piwowar-Manning E,
Shrestha R, Redd AD, Rothman RE, Sullivan D, Shoham S, Casadevall A, Quinn TC, Pekosz
204 A, Tobian AAR, Laeyendecker O. 2021. Comparative Performance of Five Commercially
Available Serologic Assays To Detect Antibodies to SARS-CoV-2 and Identify Individuals
with High Neutralizing Titers. *J Clin Microbiol* 59.
- 15. Di Germanio C, Simmons G, Thorbrogger C, Martinelli R, Stone M, Gniadek T, Busch MP.
2022. Vaccination of COVID-19 convalescent plasma donors increases binding and
neutralizing antibodies against SARS-CoV-2 variants. *Transfusion* 62:563-569.
- 16. Bloch EM, Focosi D, Shoham S, Senefeld J, Tobian AAR, Baden LR, Tiberghien P, Sullivan
DJ, Cohn C, Dioverti V, Henderson JP, So-Osman C, Juskewitch JE, Razonable RR, Franchini
212 M, Goel R, Grossman BJ, Casadevall A, Joyner MJ, Avery RK, Pirofski L-a, Gebo KA. 2023.
Guidance on the Use of Convalescent Plasma to Treat Immunocompromised Patients With
Coronavirus Disease 2019 (COVID-19). *Clinical Infectious Diseases* doi:10.1093/cid/ciad066.

**Funding-** Bloomberg Family Foundation

**Authorship** Contribution: DS & MC designed data analysis from existing hospital data. MC analyzed
data, DS drafted manuscript with input from all (OL and AT) authors and all authors approved the final
version of the manuscript.

**Conflict-of-interest disclosure:** All authors report no relevant disclosures.

Figure legends

**Figure 1. Increasing trend of spike IgG antibodies in a tertiary hospital patient population.** The 3-
224 year period between April 2020 to February 2023 was divided into five time periods to represent the
225 alpha variant (up to January 2021) the alpha variants with partial vaccinations (to June 2021), the delta
variants (to December 2021), the omicron BA.1 and 2 (to July 2022). and the omicron BA.4/5 with
BQ.1 and XBB variants (to January 2023). A total of 15,820 sera are shown by the individual points,
contributed by 11,022 patients. Most patients (8,451, 77%) were tested only once during the 3 years,
while the remaining 2,571 underwent sequential measurements. The box plot in each of the five time
periods represent the middle 50% of the observations, bordered at the 25th and 75th percentiles, and
contain a line indicating the median antibody value. The 3.5 ELISA optical density value suggested by
the FDA as cutoff for COVID-19 convalescent plasma donations is shown by the red dashed line.

**Figure 2A. Sequential spike IgG antibody levels in 49 of the total 2,571 patients with sequential**
**measurements.** The 49 patients were tested between 7 and 13 times, contributed 381 sera, and had a
mean follow-up time of 381 days. The shaded area represents the 95% confidence interval around the
linear fit (which is not shown).

**Figure 2B. Comparison of the Euroimmun spike antibody ELISA assays, the original version**
**released in May 2020 and the newer version with a standard curve released in October 2021.** The
dotted lines indicated the FDA recommended threshold for COVID-19 convalescent plasma donation:
3.5 AU for the original assay on y-axis and 55 RU/mL for the newer assay on x-axis. The comparison
was made in June 2021 for the grey and February 2023 for the red symbols.

Dr. Bar-On Yotam
Editor, Microbiology Spectrum
ybaron@technion.ac.il

Dear Dr. Bar-On Yotam,

Thank you very much for the effort you and the reviewers dedicated to our paper [Spectrum 02183-23]. We were pleased to read that only minor modifications are required before publication. The two reviewers made several insightful comments, which have helped us to improve the paper. Our point-by-point rebuttal is presented here below. The corresponding changes in the text are also indicated.

From Reviewer 1

1. *The study could have included virus neutralization tests in a subset of patients from each of the time intervals, as to correlate the total serum antibody levels with their neutralization function.*

The point is well taken. However, it is well-established that antibody concentrations positively correlate with neutralization function. For example, Otter et al in Microbiology Spectrum compared 138 convalescent blood donor sera using the same Euroimmun assay we used to measure total antibody levels and a neutralization assay. The authors reported a highly significant positive correlation (Pearson r coefficient of 0.83, $p < 0.0001$) [Figure 5 in (1)]. Numerous other studies have described increases in neutralization titer over the course of the pandemic in association with increased total antibody levels. For example: a 9-fold increase from the 311 geometric mean titer from 27 studies against the WA-1 strain to the 2753 titer after both pre-omicron COVID-19 and vaccination in 19 studies; and a 10-fold rise (from 15 to 192) with Omicron BA.1 tested in parallel in the same convalescent plasma studies (2). Convalescent plasma after vaccination in 12 studies harvested in 2022 during Omicron has GMT neutralizations near 6000 essentially a 20-fold increase(3). Finally, it is important to remember the design of the present study: our was a retrospective study aimed to analyze the total spike IgG levels measured in a routine clinical laboratory, a laboratory not equipped with the research tools that are needed to perform viral neutralization assays.

2. *The authors could have enriched the study by associating the spike antibody results with other variables such as clinical, epidemiological, or biological variables of the virus.*

Thank you for the suggestion. Routine clinical laboratories do not have access to epidemiological data or biological characteristics of the virus, but can integrate laboratory results with basic clinical characteristics, such as age and gender. We have integrated the reviewer's suggestion in the revised manuscript, as detailed in our responses to the specific suggestions made by the reviewer.

3. *The authors could add a table (plot) describing the population analyzed. This could be organized by time intervals; in each of them, it could be indicated, for example, the number of women and men, the number of people vaccinated, the number of people with previous SARS-CoV-2 infections, and the number of seropositive patients. This table could be cited in the text.*

We have now added a table (Supplemental Table 3) that captures the contemporaneous cumulative SARS-CoV-2 cases in the State of Maryland, along with the percentage of population vaccinated, seroprevalence from CDC Seronet, and the CDC Tracker data from the blood donor seroprevalence studies. Corresponding changes have been integrated in the manuscript.

4. *In addition to the median, it can be informative to add to the text the minimum and maximum values of the number of antibodies.*

We have now calculated median, minimum and maximum for the five COVID-19 time periods and presented the results in a new table (Supplemental Table 2).

5. *The authors could evaluate if there are significant statistical differences between the median number of antibodies of the different time intervals.*

Thank you for the suggestion. Spike antibody levels increase significantly from one period to the next one, both in a crude (unadjusted) analysis and after adjusting for gender and age. We have presented these new results in the revised text and in Supplemental Table 2.

6. *The authors could evaluate if there are significant statistical differences in the number of antibodies between sexes for the same time interval or between different intervals for the same sex.*

We have now performed this statistical analysis. Gender was indeed strongly associated with spike antibody levels, in keeping with the notion that the number of B lymphocytes and the levels of circulating antibodies are higher in women (4). We have integrated these new findings in the revised text and in Supplemental Table 1.

From Reviewer 2

We thank the reviewer for considering our study interesting and having important implications for the use of COVID-19 convalescent plasma therapy.

1. *The authors mention that the FDA uses a threshold of 3.5 AU to qualify CCP donors. However, they do not discuss why this threshold was chosen and why the 55 AU was also chosen in the newer assay.*

The FDA first issued the Emergency Use Authorization (EUA) for administering CCP to hospitalized patients in August 2020, qualifying donors based on a documented history of COVID-19 or a positive spike antibodies Orthos test. The EUA was later revised in Feb 2021 to include several commercial spike antibody assays, such as the Euroimmun assay used in this study. In this assay, the positivity threshold recommended by the manufacturer is 1.1 ELISA optical density (OD) ratio (the ratio derived by dividing the optical density obtained with the patient serum by the density obtained with the single calibrator provided with the kit). The positivity threshold validated in our laboratory was slightly higher, at 1.23 (5); and the qualification threshold for CCP donors chosen by the FDA was 3.5. About one year later, in February 2022, the FDA authorized the use of a newer version of the Euroimmun assay, called QuantiVac. This newer version features 6 calibrators (rather than just one), thus allowing the derivation of a standard curve and the expression of results as arbitrary, relative units per mL (rather than as a ratio of optical densities). The FDA selected a value of 55 RU/mL as the qualification threshold for CCP donations, thus suggestion of conversion factor of 15.7 between the two Euroimmun assays (that is, a value of 1 OD ratio in the original assay corresponds to a value of 15.7 RU/mL in the QuantiVac assay). In this study (Figure 2B), we showed that the conversion factor between the two assays is, in our hands, 14, thus similar to the 15.7 in the FDA data. But the bottom line is that the qualification threshold for CCP donations selected by the FDA when using

Euroimmun assays (either the 3.5 of the original assay or the corresponding 55 in the newer assay) has not essentially changed over the past 2 years. Our data (Figure 1) show that while using these thresholds would have qualified only the top 15% of donors during the initial period (March 2020 – December 2020), they would have qualified almost everybody as donor (95%) during the last period (July 2022 – February 2023). This selection strategy is detrimental to recipients of CCP because studies have shown that the benefits of CCP transfusions are greater when the levels of spike antibodies they contain are higher (see point 2 below for references).

2. *The authors suggested that restricting CCP donors to those with high titer spike antibodies may be more effective in protecting immunocompromised patients from variants. However, they do not provide any data to support this suggestion. It would be helpful to see data on the clinical effectiveness of CCP therapy in patients with different levels of spike antibodies.*

Among outpatients with COVID-19, performing a meta-analysis of 5 randomized trials 2,620 patients, Levine and colleagues reported that CCP was more effective in reducing all-cause hospitalization when given within 5 days of symptom onset and contained high (top half of research donor units) antibody titers. (6). Similarly, in hospitalized patients, Joyner and colleagues noted reductions in death when using CCP of higher antibody levels (7). A large meta-analysis in hospitalized patients noted better outcomes with higher antibody doses in those already hospitalized (8). Overall, data suggest that is best to administer CCP that contains high antibody titers. As far as the immunosuppressed population is concerned, clinical data are sparse during the period of this study, so we have revised the final statement of the Abstract accordingly.

3. *Abstract, line 34, yellow highlight: (6,362 females, mean age 50±21 years)*

We have now revised this line by specifying the numbers in both genders.

4. *Abstract, lines 41-43, yellow highlight*

We have now revised the English to make the conclusion clearer.

5. *Importance, line 51, yellow highlight*

We have now removed the word limit.

6. *Text, lines 85-86, yellow highlight*

We have changed the text to “who were tested longitudinally two or more times”.

7. *Text, lines 101-103, yellow highlight*

We have now better explained the spike antibody thresholds the FDA had chosen to consider donors eligible for CCP donations.

8. *Text, lines 111-112, yellow highlight*

We have replaced “upper quintile” with “upper decile” (also throughout text).

9. *Text, line 15, yellow highlight (“this sentence is not really clear”).*

We have deleted this sentence in the revision.

10. Text, lines 130-131 and 136, yellow highlight.

This paragraph has been largely re-written.

References cited in this rebuttal

1. Otter AD, Bown A, D'Arcangelo S, Bailey D, Semper A, Hewson J, Catton M, Perumal P, Sweed A, McKee DF, Jones J, Harvala H, Lamikanra A, Zambon M, Andrews N, Whitaker H, Linley E, Mentzer AJ, Skelly D, Knight JC, Klenerman P, group UPEt, Amirthalingam G, Taylor S, Rowe C, Vipond R, Brooks T. 2022. Implementation and Extended Evaluation of the Euroimmun Anti-SARS-CoV-2 IgG Assay and Its Contribution to the United Kingdom's COVID-19 Public Health Response. *Microbiol Spectr* 10:e0228921.
2. Sullivan D, Franchini M, Joyner M, Casadevall A, Focosi D. 2022. Analysis of anti-SARS-CoV2 Omicron neutralizing antibody titers in different vaccinated and unvaccinated convalescent plasma sources. *Nature Communications* NCOMMS-22-18877.
3. Sullivan DJ, Franchini M, Senefeld JW, Joyner MJ, Casadevall A, Focosi D. 2023. Plasma after both SARS-CoV-2 boosted vaccination and COVID-19 potently neutralizes BQ.1.1 and XBB.1. *J Gen Virol* 104.
4. Klein SL, Flanagan KL. 2016. Sex differences in immune responses. *Nat Rev Immunol* 16:626-38.
5. Caturegli G, Materi J, Howard BM, Caturegli P. 2020. Clinical Validity of Serum Antibodies to SARS-CoV-2 : A Case-Control Study. *Ann Intern Med* 173:614-622.
6. Levine AC, Fukuta Y, Huaman MA, Ou J, Meisenberg BR, Patel B, Paxton JH, Hanley DF, Rijnders BJ, Gharbharan A, Rokx C, Zwaginga JJ, Alemany A, Mitjà O, Ouchi D, Millat-Martinez P, Durkalski-Mauldin V, Korley FK, Dumont LJ, Callaway CW, Libster R, Marc GP, Wappner D, Esteban I, Polack F, Sullivan DJ. 2023. COVID-19 Convalescent Plasma Outpatient Therapy to Prevent Outpatient Hospitalization: A Meta-analysis of Individual Participant Data From Five Randomized Trials. *Clinical Infectious Diseases* doi:10.1093/cid/ciad088.
7. Joyner MJ, Carter RE, Senefeld JW, Klassen SA, Mills JR, Johnson PW, Theel ES, Wiggins CC, Bruno KA, Klompas AM, Lesser ER, Kunze KL, Sexton MA, Diaz Soto JC, Baker SE, Shepherd JRA, van Helmond N, Verdun NC, Marks P, van Buskirk CM, Winters JL, Stubbs JR, Rea RF, Hodge DO, Herasevich V, Whelan ER, Clayburn AJ, Larson KF, Ripoll JG, Andersen KJ, Buras MR, Vogt MNP, Dennis JJ, Regimbald RJ, Bauer PR, Blair JE, Paneth NS, Fairweather D, Wright RS, Casadevall A. 2021. Convalescent Plasma Antibody Levels and the Risk of Death from Covid-19. *N Engl J Med* 384:1015-1027.
8. Senefeld JW, Gorman EK, Johnson PW, Moir ME, Klassen SA, Carter RE, Paneth NS, Sullivan DJ, Morkeberg OH, Wright RS, Fairweather D, Bruno KA, Shoham S, Bloch EM, Focosi D, Henderson JP, Juskewitch JE, Pirofski L-a, Grossman BJ, Tobian AAR, Franchini M, Ganesh R, Hurt RT, Kay NE, Parikh SA, Baker SE, Buchholtz ZA, Buras MR, Clayburn AJ, Dennis JJ, Diaz Soto JC, Herasevich V, Klompas AM, Kunze KL, Larson KF, Mills JR, Regimbald RJ, Ripoll JG, Sexton MA, Shepherd JRA, Stubbs JR, Theel ES, van Buskirk CM, van Helmond N, Vogt MNP, Whelan ER, Wiggins CC, Winters JL, Casadevall A, Joyner MJ. 2023. Mortality rates among hospitalized patients with COVID-19 treated with convalescent plasma A Systematic review and meta-analysis. medRxiv doi:10.1101/2023.01.11.23284347:2023.01.11.23284347.

August 29, 2023

Dr. David J Sullivan
Johns Hopkins University Bloomberg School of Public Health
Molecular Microbiology & Immunology
615 N. Wolfe St., Rm. W 4606
Baltimore, MD 21205

Re: Spectrum02183-23R1 (Hundred-fold increase in SARS-CoV-2 spike antibody levels over three years in a hospital clinical laboratory.)

Dear Dr. David J Sullivan:

Your manuscript has been accepted, and I am forwarding it to the ASM Journals Department for publication. You will be notified when your proofs are ready to be viewed.

Sincerely,

Bar-On Yotam
Editor, Microbiology Spectrum
